# Polymer Nanocomposite Ultrafiltration Membranes: The Influence of Polymeric Additive, Dispersion Quality and Particle Modification on the Integration of Zinc Oxide Nanoparticles into Polyvinylidene Difluoride Membranes

**DOI:** 10.3390/membranes10090197

**Published:** 2020-08-24

**Authors:** Thorsten van den Berg, Mathias Ulbricht

**Affiliations:** 1Lehrstuhl für Technische Chemie II, Universität Duisburg-Essen, 45141 Essen, Germany; thorsten.vandenberg@gmail.com; 2Center for Nanointegration Duisburg-Essen (CENIDE), 47057 Duisburg, Germany

**Keywords:** nanocomposite, ultrafiltration membrane, nanofiller dispersion, phase separation, PVDF, ZnO

## Abstract

This study aims to improve the understanding of the influence of metal oxide nanofillers on polyvinylidene difluoride (PVDF) ultrafiltration membranes. Zinc oxide nanoparticles were chosen as the model filler material. The membranes were prepared by non-solvent induced phase separation from PVDF solutions in *N*-methylpyrrolidone. The influences of the addition of polyvinylpyrrolidone (PVP), the nanoparticle dispersion quality, and a surface modification of the ZnO particles with PVP on the nanofiller integration into the polymer matrix and the resulting membrane separation performance, were evaluated. Unmodified and PVP-modified nanoparticles were characterized by evaluation of their Hansen solubility parameters. The membranes were characterized by ultrafiltration experiments, scanning electron microscopy (SEM) and with respect to mechanical properties, while the dope solutions were analyzed by rheology in order to judge about dispersion quality. Pure water permeability and solute rejection data revealed that the dominant effect of the addition of pristine ZnO nanoparticles was a major decrease in permeability caused by pore blocking. In SEM analyses, it was seen that the plain nanofiller did not integrate well into the polymer matrix. Importantly, it was found that the surface modification of the nanofiller, as well as a high dispersion quality, can be strategically used to enhance the integration of the nanofiller and thus suppress pore blocking, leading to membranes with high ultrafiltration rejection and permeability simultaneously. Overall, the study provides relevant insights into a new approach to integrating nanofillers into polymer nanocomposite membranes for improving their properties and performance.

## 1. Introduction

Polymers are the main material for the preparation of state-of-the-art ultrafiltration (UF) membranes because they offer a wide range of attractive properties, and membrane preparation by various methods is easily scalable [1,2,3]. Polyvinylidene difluoride (PVDF) is widely used as a base polymer for ultrafiltration membranes; highly porous membranes with tunable barrier properties and excellent stability against mechanical, thermal and chemical stresses can be obtained with it. The main disadvantage related to typical UF applications is the hydrophobicity of the material. This makes the membrane prone to fouling, and leads to a reduction in the membrane performance by reducing the permeability and additional costs for membrane cleaning or replacement [3].

Many efforts have been made toward increasing the hydrophilicity of PVDF membranes to overcome this problem [4,5,6,7]. Two general approaches can be distinguished. On the one hand, surface modification of already established membranes encompasses the generation or introduction of hydrophilic groups onto the surface of the membrane material, e.g., by plasma treatment or electron beam irradiation, or by surface selective coating or polymerization. On the other hand, blend modification encompasses the addition of hydrophilic additives to the dope solution during the membrane preparation. The additive may be enriched on the surface but must also be anchored in the bulk of the membrane; therefore, it can affect the corresponding properties of both membrane bulk and surface. Hydrophilic polymers, such as polyvinylpyrrolidone (PVP) or polyethylene glycol (PEG), are employed as additives in industrial PVDF UF membrane manufacturing.

More recently, inorganic oxides are also being intensively considered as filler material, e.g., titania, zirconia, or zinc oxide [8,9,10]. With the advent of nanotechnology, the interest shifted toward the use of nanoparticles as filler material. The size of nanofiller is significantly smaller than that of conventional filler materials, and therefore nanofillers can also be immobilized on the polymer surface during membrane preparation. Thus, the contribution of the nanomaterial’s surface to the surface structure of the membrane can affect hydrophilicity, charge, and the exposure of specific functional groups. When the nanofiller particle has photocatalytic, adsorptive, or anti-bacterial properties, these are added to the nanocomposite membrane in the same way. An increase in the surface roughness is also often observed. In parallel, the bulk properties of the membranes, especially the resistance against mechanical stress, can also be improved by the addition of nanofillers.

The main problem is the agglomeration of the nanofiller during the phase separation, which causes a reduction in modification effectiveness. The main reason is an incompatibility between the polymeric matrix material and the nanomaterial [7]. Agglomeration increases the particle size; this reduces the overall surface area of the particles and the effective particle concentration, and both can impair the expected effect of the nanoparticle modification. Furthermore, large agglomerates can decrease the permeability of the membrane because they can block pores and therefore hinder convective flux through the membrane [10,11].

Regarding the influence of metal oxide nanofillers on the performance of PVDF ultrafiltration membranes, 18 representative studies were identified in which the membranes were prepared by non-solvent induced phase separation (NIPS). The nanoparticles employed in these studies were silica, titanium oxide, aluminum oxide, magnesium hydroxide and zinc oxide. Since these are all metal oxides, it is expected that their effect on the PVDF membrane’s formation will be similar to the effect of zinc oxide because of similar hydrophilicity and Hansen solubility parameters [12]. An overview table, showing the most important results of these studies [11,13,14,15,16,17,18,19,20,21,22,23,24,25,26,27,28], is provided in the Appendix A (Appendix A), and the most important results are recapitulated in the following paragraphs.

With regard to the permeability, an enhancement was typically seen as a consequence of the addition of nanoparticles to the dope solution used for membrane preparation. This was demonstrated in 14 studies, and in some of the cases this was attributed to the measured increase in surface hydrophilicity caused by the hydrophilic particles [13,14,15,16,17,18,19,20,21,22,23,24,25,26]. Such increase in hydrophilicity was also reported in many cases [13,14,15,16,17,18,19,20,21,23,24]. In seven studies, the permeability decreased at higher nanoparticle concentrations; this was attributed to pore blocking by agglomerated nanofiller [14,15,16,17,18,23,25]. That the addition of nanoparticles solely led to a decrease in permeability was reported in three studies [11,27,28], while two of these studies also reported an increase in hydrophilicity [11,28]. Overall, the different influence of the metal oxide nanofiller on permeability in different studies could not be correlated with one or more obvious factors during the membrane preparation via NIPS.

With regard to the membrane structure, the assessment of the effect of nanoparticles was rather inconclusive. No effect was seen in four studies [11,13,23,24]. An increase in skin layer thickness was observed in two studies [18,25]. The authors attributed this to the increased viscosity of the dope solution, which was caused by the addition of the nanoparticles. The macro-void formation in the membrane cross-section was decreased in five studies, likely caused by the same change in viscosity [14,16,18,25,28]. In contrast, in three other studies an increase in macro-void formation tendency with nanofiller addition was observed [17,20,21], and one study showed an initial increase in macro-void formation that deceased at higher particle concentration [15].

The mechanical properties of the membranes were evaluated in some of the studies. The incorporation of the nanoparticles was found to increase the tensile strength with a decrease of elongation at break in one study [26]. A similar result, but without looking into the elongation at break, was also reported elsewhere [21]. The decrease of both properties was observed in another study [18]. However, in four other studies an increase of both tensile strength and elongation at break was observed [13,14,20,23].

Overall, contradictory results and interpretations have been reported in the literature about the effect of metal oxide nanofillers on the structure and separation performance of PVDF-based ultrafiltration membranes. One deficit of all of the previous studies is the lack of a quantitative characterization of the quality of the dispersion of the nanofiller in the casting solution used for NIPS-based membrane formation. As such, the primary objective of this study was to evaluate in detail the influence of metal oxide nanofillers, here ZnO, on the preparation of PVDF ultrafiltration membranes via NIPS, with particular emphasis on the influence of the hydrophilic additive PVP and the nanofiller dispersion quality. PVP is extensively used in nanoscience to increase the longevity of nanoparticle dispersions via steric stabilization [29]. The hydrophilic polymer can form hydrogen bonds with the metal centers of oxides via its carbonyl group leading to strong physisorption [30]. For this reason, it was decided to also study the influence of a simple surface modification of the ZnO nanoparticles with PVP on nanofiller dispersion quality, and the resulting PVDF membrane structure and performance. To the author’s knowledge, such nanofiller surface modification had not before been reported in conjunction with polymer membrane fabrication via NIPS.

## 2. Materials and Methods

### 2.1. Materials

The membrane polymer was PVDF Solef 6010 from Solvay Specialty Polymers (Brussels, Belgium). The polymeric additive was PVP Luvitec K-30 from BASF, a low molecular weight type specified with a viscosity-derived K value of 30. Zink oxide nanoparticles were type VP ZnO 20 from Evonik Industries with a primary particle size of 20 nm. An overview of all the materials used in this study, including quality/purity as well as supplier/manufacturer can be found in Appendix A, Appendix A. The deionized water (DI water) used in all experiments was purified with a Milli-Q system from Merck-Millipore, Darmstadt, Germany.

### 2.2. Preparation of Nanoparticle Dispersion 

The nanoparticle dispersions were prepared using sonication with a sonotrode (Sonoplus HD 3200 from Bandelin Electronics GmbH & Co. KG, Berlin, Germany). Rosette cells were used to facilitate axial mixing. Two sonotrode tips were used in this work, i.e., *VT 70* for volumes in excess of 50 mL and *KE 73* for smaller volumes. The sonication was commenced for 10 min at an amplitude of 50% with the cells in a cooling bath of ice and water.

### 2.3. Nanoparticle Surface Modification 

First, PVP was dissolved in ultrapure water under constant stirring. Then, pristine ZnO nanoparticles are added to the PVP solution and the dispersion was prepared by sonication as described in Section 2.2. The particles were separated from the aqueous phase by centrifugation (using a Universal 320 from Hettich Lab Technology, Tuttlingen, Germany) at 4500 rpm. The sediment was rinsed with ethanol twice and then dried in a vacuum oven at 40 °C.

### 2.4. Dynamic Light Scattering for Analysis of Nanoparticle Dispersions 

The Stabisizer PMX 200C from Particle Metrix (Meerbusch, Germany) was used for the determination of the hydrodynamic particle diameter in the dispersions. For each dispersion three measurements over 90 s were conducted. For the calculation, the ZnO particles were assumed to be of spherical shape with a refractive index of n = 2.03 (at 780 nm) [31]. The mean value from the three measurements and standard deviation were calculated from the size distribution by number.

### 2.5. Hansen Solubility Parameter Evaluation for Nanoparticles 

As the first step, dispersions of the nanoparticles were prepared in the selected dispersants with widely different solvent properties. The mass fraction of particles in the solvent was kept at 0.1% *w/v* and the conditions for dispersing by sonication were as stated in Section 2.2. The mean hydrodynamic particle size (d_H_) of the dispersion was characterized by dynamic light scattering (see Section 2.4) immediately after preparation and then once each day over the next four days or until the dispersion became unstable. Instability is reached when most particles are sedimented. The stability of the dispersion was rated according to the criteria listed in Table 1. As the second step, the calculation of the Hansen solubility parameter of the nanoparticles was performed with the software HSPiP 4.1 (www.hansen-solubility.com), using the standard sphere algorithm without consideration of the molar volume and no limitation of the interaction radius.

### 2.6. Stability of the Surface Modification of Nanoparticles

The following experiment was conducted to evaluate if the surface modification of the nanoparticles with PVP (cf. Section 2.3) can persist through the preparation of the dope solution as described in Section 2.8. For this 0.5 g of the PVP-modified particles were dispersed in 30 g N-methylpyrrolidone (NMP) under standard conditions (see Section 2.2). The dispersion was then stirred at 60 °C for two days to simulate the step conducted to dissolve the polymer. The particles were then removed from the dispersion by centrifugation and rinsed with ethanol twice. Thereafter, the particles were dried in a vacuum oven at 40 °C. The particles were then characterized by thermogravimetry (see Section 2.7) and IR spectroscopy (FT-IR Varian 3100 Excalibur, Varian Inc., Palo Alto, CA, USA). The same procedure was also performed with pristine ZnO particles as reference.

### 2.7. Thermogravimetric Analysis

The thermogravimetric analysis of the nanoparticles was performed on the STA449 F3 Jupiter from NETZSCH (Selb, Germany). Between 15 and 40 mg of the sample were placed in an alumina crucible. The analysis was performed in the range of 30 to 800 °C. The heating rate was set to 10 K/min and an argon stream of 20 mL/min was present at all stages of the measurement.

### 2.8. Dope Solution Preparation

All dope solutions consisted of 16% PVDF in NMP and also contained in some cases 1% PVP. The polymers were dried at 40 °C in the vacuum oven for 72 h before use. Quantities of 8 g PVDF and, if applicable, 0.5 g PVP were weighed into an Erlenmeyer flask with glass joint and a stir bar was added. The flask was closed with the plug, sealed with Parafilm and placed on a magnetic stirrer hot plate. If nanoparticles were to be added, the particles were dispersed in the NMP as described in Section 2.2. The 41.5 g NMP with or without dispersed particles is then added to the polymer(s) under stirring at 300 rpm. The flask was then again closed and sealed. The solution was stirred at 300 rpm for 48 h at 60 °C to assure that the polymer(s) were well dissolved. Afterwards, the dope solution was allowed to rest for a few hours in the dark at room temperature to remove bubbles.

### 2.9. Rheology

The shear viscosity of the dope solution was measured using a rheometer Physica MCR301 from Anton Paar (Graz, Austria). The measurement was performed using a probe with conical geometry and diameter of 25 mm with the designation CP25-2. The sample was kept under a dry air stream during the measurement to prevent precipitation over the course of the measurement. The measurement cell was kept at 20 °C. The shear viscosity was recorded over the shear rate from 1 to 800 s^−1^. The viscosity measured at 15.9 s^−1^ (also corresponding to the shear rate applied during membrane casting; cf. Section 2.10) was selected to compare the dope solutions.

### 2.10. Membrane Casting

The membrane preparation was conducted in a climate box that kept the relative humidity below 30% RH at ambient temperature. The membranes were casted using a COATMASTER (Erichsen GmbH, Hemer, Germany). The casting speed was fixed at 5 mm/s. A casting knife with a clearance of 200 µm was used. The casted dope solution film was immediately immersed in the coagulation bath outside of the climate box. The coagulation bath consisted of DI water. The membrane remained in the coagulation bath for one hour and was then rinsed in a fresh DI water bath for 24 h. For the ultrafiltration experiments (Section 2.11), the samples were taken from the wet membrane sheet and stored in 10 mM aqueous sodium azide solution. The rest of the membrane was immersed in ethanol overnight and then the solvent was exchanged to *n*-hexane. The membrane was then dried from *n*-hexane and used for SEM characterization (see Section 2.13).

### 2.11. Membrane Performance Evaluation by Crossflow Ultrafiltration 

The crossflow filtration setup was a laboratory device LSta05 from SIMA-tec GmbH (Schwalmtal, Germany). The overall area of the flat membrane sample was 100 cm², while effectively 84 cm² were used in the filtration. After installing the membrane in the cell, the feed container was filled with two liters of aqueous sodium chloride solution (c = 50 mg/L) for compacting and pure water permeability. The volume flow was kept at 20 L/h for the whole experiment. The pure water permeability was measured at 0.5 bar after compacting the sample at 1.5 bar for 30 min. The device gave a continuous reading of flux and trans-membrane pressure. The mean permeability was calculated from this data for each stage of the experiment. The ultrafiltration of polyethyleneglycol (PEG) 35 kDa and polyethyleneoxide (PEO) 100 kDa was performed at a pressure of 0.15 bar; the feed concentration was set to 1 g/L. The first samples of the feed and permeate were taken 30 min after reaching constant conditions with respect to flux and a second sample was taken after another 30 min. The solute concentrations in feed (c_F_) and permeate (c_P_) were determined by analysis of total organic carbon (TOC) (see Section 2.12). The rejection was calculated by Equation (1).
(1)$$R=\left(1-\frac{c_{P}}{c_{F}}\right)\times 100\%$$

### 2.12. Total Organic Carbon Analysis 

The fraction of total organic carbon (TOC) was determined using the instrument TOC-V CPN from Shimadzu (Duisburg, Germany), which was equipped with the external autosampler ASI-V. The device was a combustion analyzer and was calibrated for analyses in the range of 0–1000 mg/L TOC. The method can tolerate up to 10 mg/L inorganic carbon. Two milliliters of the sample were diluted with 15 mL ultrapure water to stay within the calibration range.

### 2.13. Scanning Electron Microscopy 

The membrane morphology was analyzed by scanning electron microscopy. The instrument used was the ESEM Quanta 400 FEG from FEI (ThermoFisher Scientific, Hillsboro, OR, USA). The dry samples were quenched in liquid nitrogen and then broken in half to obtain also samples which are suitable for the analysis of cross-section morphology. The membrane samples were sputtered with gold/palladium (80/20) to ensure sufficient surface conductivity.

### 2.14. Mechanical Characterization 

The mechanical properties of the membranes were characterized regarding tensile strength, Young’s modulus, and elongation at break. The tests were performed on the material testing instrument Test Expert^®^ II from Zwick Roell (Ulm, Germany). Beforehand, the samples were cut into rectangular pieces with a 2 cm width and 10 cm length. The samples were then pre-stressed at 2 N with a testing speed of 100 mm/min in standard climate, i.e., at 25 °C and 65% RH. Five samples of each membrane were subsequently tested and the mean value and standard deviation are presented in this work.

## 3. Results and Discussion

### 3.1. Influences of the Polymeric Additive Polyvinylpyrrolidone and of the Nanofiller Zinc Oxide

PVDF dope solutions in NMP, containing different fractions of ZnO nanoparticles, were prepared with and without the addition of 1% PVP according to the procedure described in Section 2.8. All solutions were characterized by rheology; the results are shown in Figure 1. First, the viscosity of the PVP-containing solutions was considerably higher than those without; this is the well-known effect of the polymeric additive, which can be used to modulate membrane formation via NIPS and thus tune UF membrane performance [2,3,4]. Second, both types of dope solutions exhibited a steady increase in viscosity with the addition of ZnO nanoparticles. The extent of the increase was the same for both types of dope solutions. This steady increase indicates that the particles are well dispersed, because the magnitude of the interactions between nanoparticle surface and solvent rose proportionally with nanoparticle fraction [32]. This also indicates that the addition of PVP does not influence the quality of nanoparticle dispersion in the dope solution.

Membranes were prepared from both types of dope solutions. An overview of the ultrafiltration performance of the membranes containing 0% and 50% *w/w* (PVDF) zinc oxide nanoparticles is displayed in Table 2.

The addition of PVP is known to predominately increase the barrier layer porosity of the membrane [33,34], which is here manifested by the much higher permeability and, in the case of the small test solute (PEG 35 kDa), a lower solute rejection compared to the membranes prepared without PVP. This effect is also directly seen in the density of surface pores, as displayed in the SEM images of the outer membrane surface (Figure 2a,b).

The addition of the nanoparticles reduces the permeability, while the solute rejection is increased (for PEG 35 kDa; cf. Table 2). Agglomerated nanoparticles are seen in the SEM images of the outer membrane surface (Figure 2c,d). These particles are located below the skin layer and seem to shine through it because of the higher conductivity of ZnO compared to the organic polymer. It can be concluded that the observed decrease in permeability is caused by these agglomerates leading to pore blocking, since the surface porosity of the polymer matrix seems not to be affected. The membrane surfaces in Figure 2a,c exhibit a low density of surface pores. This indicates that the hydrophilic zinc oxide nanoparticles exhibit no porogenic effect. The porogenic effect of hydrophilic compounds can be separated into a kinetic and a thermodynamic component. The kinetic effect is associated with an increase in viscosity, as was seen in this series. The thermodynamic effect is the delaying of the demixing, which can be determined by cloud point experiments to assess the fraction of coagulant required for different dope solutions. It was not possible to conduct these as the dope solutions were too opaque for a measurement with sufficient sensitivity.

The cross-section morphology for both membrane types with nanofiller is displayed in Figure 3. When comparing the images, it is obvious that the agglomerate size is larger in the membrane prepared without the addition of PVP. This can be related to the higher viscosity of the dope solution with PVP (cf. Figure 1). The rheology data also indicated that the particles in both types of dope solution exhibit a comparable degree of dispersion. Therefore, the agglomeration of the particles must have occurred during the phase separation process. For that, the particles must have moved by diffusion to form agglomerates. The higher viscosity decreases the diffusion rate and thus limits the size of the resulting agglomerates in the membrane prepared from the dope solution with PVP. Hence, it is hypothesized that different sizes of ZnO agglomerates are fixed upon the solidification of the PVDF during the NIPS process, depending on the viscosity of the dope solution modulated by addition of PVP (cf. also data and further discussion in Section 3.2).

Furthermore, all agglomerates seen in Figure 3—either the larger ones obtained without PVP or the smaller ones in the presence of PVP—are positioned just at the pore walls of the macro-voids. Therefore, it may be concluded that the addition of PVP does not have a direct influence on the integration of the particles into the polymer matrix.

### 3.2. Influence of Zinc Oxide Nanofiller Dispersion Quality

A series of dope solutions without PVP was prepared by omitting the sonication of the dispersion of ZnO in NMP before using it to dissolve the membrane polymer (cf. Section 2.8). These solutions exhibited a reduced viscosity compared to the reference solutions, i.e., the ones without PVP additive shown and discussed in Section 3.1 (comparison of data is shown in Appendix A, Appendix A). The values for 20% and 50% ZnO content without sonication were about the same, while a systematic increase was observed with sonication (cf. Figure 1). The much lower viscosity at the same nanoparticle fraction for the not sonicated solutions indicates that the particles are agglomerated [32].

Membranes were prepared from the dope solutions and characterized; membrane performance data are presented in Table 3. Both membranes exhibited virtually the same water permeability and solute rejection (for the smaller test solute; a slightly higher rejection was seen for the larger test solute and the reference membrane).

The membranes were further characterized by SEM. The comparison of images of the cross-section and outer surface morphology is provided in Figure 4.

When the images above are compared, it is apparent that the reference membrane exhibits more visible and larger agglomerates over the cross-section. This seems counterintuitive, since the rheology results suggested that the particles in the dope solution of the latter membrane were agglomerated before the membrane preparation (cf. Appendix A).

However, the observed phenomena can be explained by a mechanism that is visualized in Figure 5. The initial situation can be seen under (a); the dispersed nanoparticles are distributed in the dope solution. The situation displayed under (b) shows that the dope solution has separated into polymer-rich and polymer-poor phases after immersion in the coagulation bath. The particles are still in the polymer-rich phase because the phase separation is a relatively fast process. However, the polymer has not solidified yet, so that the nanoparticles have the option to migrate toward the polymer-poor, water-containing phase. The driving force for that is the hydrophilicity of the particles; the direction of the particle movement is indicated by the arrows. In the situation displayed under (c), the polymer has precipitated and the porous membrane structure is fixed. One important differleftence between particles consisting of the same material but differing in size is their diffusivity (cf. discussion in Section 3.1). For the well-dispersed nanoparticles (cf. Figure 5, case 1), the mobility toward the phase boundaries is relatively high. Once the particles reach the polymer-poor phase, the environment consists of a mixture of NMP and water. Since the pristine zinc oxide particles are not well compatible with NMP (see Section 3.3), they tend to agglomerate. Upon solidification, some particles are enclosed in the polymer-matrix, while other nanoparticles have formed agglomerates in the void-space of the support structure. The nanoparticles which had reached the skin layer are not agglomerated since the local NMP concentration in direct contact with the coagulation bath, consisting of water, is much lower than that inside the membrane. These particles either migrate into the coagulation bath or they are removed from the skin layer during the washing of the membrane. This explains the lack of particle aggregates on the membrane surface in Figure 4a.

For the agglomerated nanoparticles (cf. Figure 5, case 2), the initial dope solution contains larger spheres. Upon phase separation, the particles have the same diffusion direction, but their diffusivity is smaller than that of the not agglomerated ones (indicated by smaller arrows). The time until solidification is virtually identical since this is predominantly a function of the polymer concentration and the polymer/solvent/non-solvent combination. Hence, the agglomerated particles travel a significantly shorter distance before the polymer matrix solidifies. Finally, the agglomerates are fixed on the outer and inner surfaces of the membrane because they cannot reach the void space or the bulk of the coagulation bath in the available time span.

This qualitative model explains the differences in agglomerate size and location observed in this study, including the effects already shown (cf. Figure 3) and discussed in Section 3.1. Furthermore, this model could also help to explain differences seen between various studies in the literature (cf. Section 1), because it emphasizes the impact of nanofiller dispersion quality, as well as that of dope solution viscosity.

### 3.3. Influence of Particle Surface Modification

Polyvinylpyrrolidone is often used as a steric stabilizer agent for nanoparticle dispersions [29]. Therefore, it is of interest to study the effect of a PVP modification of the nanoparticles on membrane preparation, structure and performance. The zinc oxide nanoparticles were modified with 500% *w/w* (ZnO) PVP (cf. Section 2.3). The characterization by thermogravimetric analysis (TGA; Figure 6) reveals that the modified particles contain 18.2% PVP on their surface. It should be noted that the TGA was conducted under a nitrogen atmosphere, which means that the polymer underwent decomposition and to some extent also carbonization, but not plain oxidation. Therefore, one can assume that the mass loss represents less than the total mass of adsorbed PVP. However, the TGA results correlate with the overall adsorbed mass of PVP and are useful as a measure of the total amount of PVP on the nanoparticle’s surface. A dispersion of these particles in NMP was stirred for two days at 60 °C to simulate the conditions present during the preparation of the dope solution (cf. Section 2.6). Unmodified zinc oxide nanoparticles were treated in the same manner as the reference. These reference particles exhibited an amount of adsorbed NMP of 1.4%, according to TGA. The TGA characterization also revealed a reduction in the PVP content of the modified particles to 5.8% (under the assumption that the amount of adsorbed NMP is the same as for the reference), as a consequence of the washing with NMP for two days at 60 °C.

The presence of PVP on the particle surface was validated by infrared spectroscopy (Figure 7). In the figure, the particles before and after washing exhibit the same signals as those attributed to PVP. The vibration of the carbonyl group can be seen at 1665 cm^−1^ and the signal of the methylene group can be seen at 1419 and 1285 cm^−1^ [35,36]. There is a decrease in the absorption after washing with NMP, which is a consequence of the partial removal of PVP from the surface of the particles, as also seen with TGA. It is, however, important to note that a significant surface modification with PVP was preserved under the conditions typically used for preparation of the dope solutions for membrane casting.

Next, the Hansen solubility parameters were evaluated for the PVP-modified and unmodified nanoparticles, following a methodology introduced earlier [37]. For that, the stability of the ZnO dispersions in various solvents was evaluated and rated according to the criteria in Table 1 (cf. Section 2.5). These primary results are shown in Table 4.

The Hansen solubility parameters were then calculated from the solvent rating; the results are presented in Table 5. The Hansen parameters of PVP and PVDF are included for comparison.

The data in Table 4 reveal that the biggest changes upon surface modification occurred for the polar aprotic solvents NMP and DMSO. While the pristine particles could not be well dispersed with these solvents (rating 6), this changed largely upon the surface modification with PVP (rating 1 for NMP, rating 2 for DMSO). The behavior of the pristine particles in NMP (“not well compatible with NMP”) had already been referred to in the interpretation of the results in Section 3.2. However, it should also be kept in mind that the preparation of the dope solutions for membrane casting involves the ultimate dispersion of the nanoparticles in a solution of PVDF in NMP, whereby the high viscosity will largely reduce the rate of agglomeration. This can explain the fact that the agglomeration of the pristine ZnO nanofiller in the PVDF membrane can be minimized by a proper protocol—including the sonication of ZnO in NMP—for dope solution preparation (cf. Section 2.8 and Section 3.1).

Looking at the results in Table 5, the tight non-covalent binding of PVP onto the nanoparticle surface increased the dispersive and polar Hansen parameters, while not affecting the Hansen parameter for hydrogen bonding. The change of the parameters of the PVP-modified particles compared to the pristine ones reveals clearly that the modification leads to a greater similarity with PVP. This is also expected to make the nanoparticles more compatible not only with the solvent NMP but also with the membrane polymer PVDF. The major change in dispersibility behavior (cf. Table 4 and Table 5) and the significant amount of residual PVP after washing the particles with NMP (cf. Figure 6) shows the strength of the hydrogen bonds between PVP and zinc oxide surfaces.

Dope solutions were prepared with the PVP-modified ZnO nanoparticles; the PVDF content was again 16%, and no additional PVP was used. The viscosities of these solutions were compared to those of analogous solutions containing unmodified ZnO particles (see Appendix A, Appendix A). The increase in viscosity was smaller for the dope solutions with the modified particles, and the discrepancy increased with increasing particle fraction. This can be explained by the presence of PVP on the particle surface. Since the solutions were prepared with equal masses of particles, the effective mass of ZnO was reduced by the amount of PVP on the particle surface. Apart from this, both solution types exhibit comparable dispersion qualities (see additional discussion in Appendix A, including Appendix A and Appendix A).

Membranes were prepared from these dope solutions. The dope solution with 50% *w/w* (PVDF) modified particles is of particular interest. The concentration of PVP desorbed from the particles into this solution should be ~1%, under the assumption that the amount of PVP desorbing during the preparation of the dope solution is the same as that seen in the experiments with NMP alone (cf. Figure 6). The total fraction of zinc oxide was 41% *w/w* (PVDF). The membrane prepared from this dope solution has therefore a comparable composition to the membranes shown and discussed in Section 3.1. The separation performances of these membranes are compared in Figure 8, while the actual data is also shown in Appendix A (Appendix A).

All the membranes exhibited a similar rejection of polyethylene oxide (M_w_ ~ 100 kDa). The rejection of polyethylene glycol (M_w_ ~ 35 kDa) was higher for the nanocomposite membranes compared to the membranes prepared only with PVP. The permeability of the membranes prepared with unmodified ZnO particles was about half of that of the membranes without particles or with PVP-modified particles. This had already been explained by the agglomeration of the unmodified particles during phase separation, which decreases permeability by pore blocking (cf. Section 3.1). In contrast, the PVP-modified ZnO particles do not exhibit this effect, and this can be related to their largely improved compatibility with NMP and PVDF (cf. above).

The cross-section morphologies of the membranes prepared with PVP-modified particles and unmodified particles are compared in Figure 9. At lower magnification (above), the agglomerates of the unmodified particles in the membrane are seen in the left image (Figure 9a); the membrane in the right image (Fig. 9b) exhibits significantly less agglomerates. This can be linked to the employed PVP-modification of the nanofiller, and it supports the conclusion that the higher permeability is caused by the suppression of pore blocking through agglomerates.

The nanofiller morphology is also compared at higher magnification; the magnified part is located on the pore wall of the macro-voids. The agglomerates of the pristine ZnO particles appear not to be integrated into the polymer matrix. In contrast, the PVP-modified particles are not easily distinguishable from the polymer matrix. This indicates a superior integration into the polymer matrix. As previously deduced from the Hansen parameter evaluation (cf. Table 5), the PVP-modification of the ZnO nanoparticles also increased their compatibility with the membrane polymer PVDF. It would have been interesting to use energy-dispersive X-ray spectroscopy mapping techniques to observe the distribution of the zinc oxide over the bulk of the membrane, but the spatial resolution in a predominantly non-conducting material was too low to provide conclusive images.

Finally, representative membranes were characterized with respect to mechanical properties, with particular emphasis on the PVDF nanocomposite membrane prepared with the PVP-modified ZnO particles. Since the pore morphology of the membrane is the feature with the greatest influence on the mechanical characterization, a high content of macro-voids decreases the stability. Therefore, membranes for this characterization were precipitated in a coagulation bath of NMP and water in a ratio of 50/50% (*v/v*), and the PVDF content in the dope solution was increased to 20%. This yields a more homogenous porous structure (SEM images can be seen in Appendix A, Appendix A). A total of 1% PVP was added to the dope solution for the reference without nanoparticles and with pristine nanoparticles in order to keep the polymer content of the samples comparable. Thus, changes in mechanical characteristics may be related to the influence of the used nanofiller on the polymer (nanocomposite) matrix. The mean values and standard deviation of the results for each membrane type are presented in Table 6.

First, it is obvious that the incorporation of pristine particles increases the tensile strength and Young modulus. This can be explained by the nanoparticles acting as physical cross-link between polymer chains, which distributes the load over more chains and increases the restoring force [39]. This indicates that at least some nanoparticles were integrated into the PVDF matrix of the membrane, while no indication could be found in the SEM images (see Section 3.1). The elongation at break was not influenced. The PVP-modified nanoparticles did not alter the tensile strength nor the Young modulus; both these values were virtually identical to those of the membrane without particles. This was very different from the effect of the pristine particles. However, the elongation at break was significantly increased by the introduction of the PVP-modified particles. This implies that the well-integrated particles create a structure that is able to endure more strain without breaking [40]. It should be noted that the standard deviation of this data is larger than that for the other membranes. This originates most likely from an inhomogeneous distribution of the particles in the membrane structure. A possible explanation for the increase in ductility is that the particles, which are well dispersed in the polymer-rich phase, reduce the tendency of the formation of crystalline domains during the phase separation. This may be caused by the nature of the modification. The particles were modified with PVP and the modification showed stability under the conditions used to prepare the dope solution. Therefore, one can expect that the particles were still covered with PVP in the finished membrane. The lack of an influence on the mechanical properties might be caused by the fact that the modified particles interact with the polymer matrix in a manner similar to PVP. The reference membrane had a similar PVP content to the membrane with PVP-modified particles, as was discussed previously.

## 4. Conclusions

It was found that the addition of hydrophilic nanoparticles into the dope solution leads to the formation of agglomerates within the PVDF membrane structure. The extent is dependent on the dispersion quality during the membrane preparation, and can be explained by the difference in diffusivity for different particle/agglomerate sizes. The proposed mechanism makes it possible to clarify the contradictory results reported for similar systems in the literature. However, a detailed analysis of the literature data, summarized in Appendix A, would still in part be speculative, especially as it is missing quantitative information on the state of the nanofiller dispersion. It should even be possible to transfer this model to other systems, and thus to increase the understanding of nanocomposite preparation. The introduction of the additive PVP to the dope solution slightly reduced the size of the nanofiller agglomerates via the increased viscosity during the phase separation. Apart from this, no additional effect was observed as a consequence of adding PVP to the dope solution. The surface modification of the particles with PVP, however, reduced the agglomerate size and improved the integration into the polymer matrix, thus avoiding pore blocking and therefore improving the membrane’s performance compared to membranes prepared with unmodified particles. This approach can be employed in membrane preparation, and enables the control of the nanoparticle distribution during the phase separation, thus allowing the preparation of a nanocomposite membrane with well-integrated and well-distributed nanofiller particles.

## Figures and Tables

**Figure 1 membranes-10-00197-f001:**
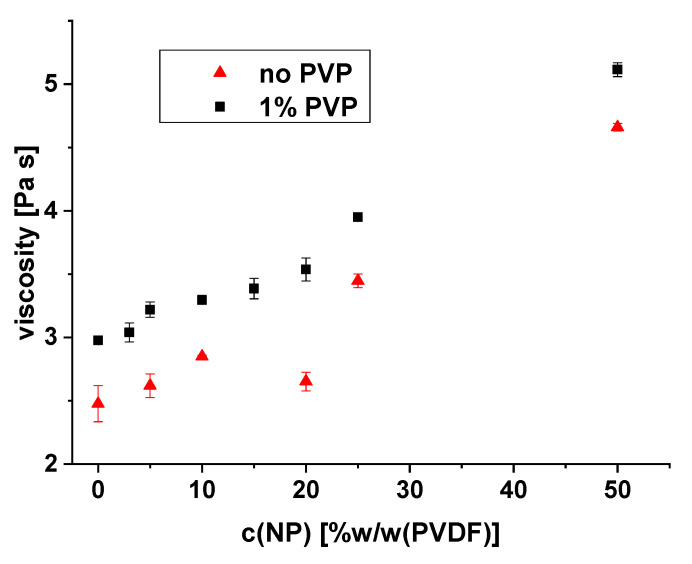
Effect of ZnO nanoparticle content on the viscosity of dope solutions consisting of PVDF in NMP, with and without PVP; measured at 15.9 s^−1^ and 20 °C.

**Figure 2 membranes-10-00197-f002:**
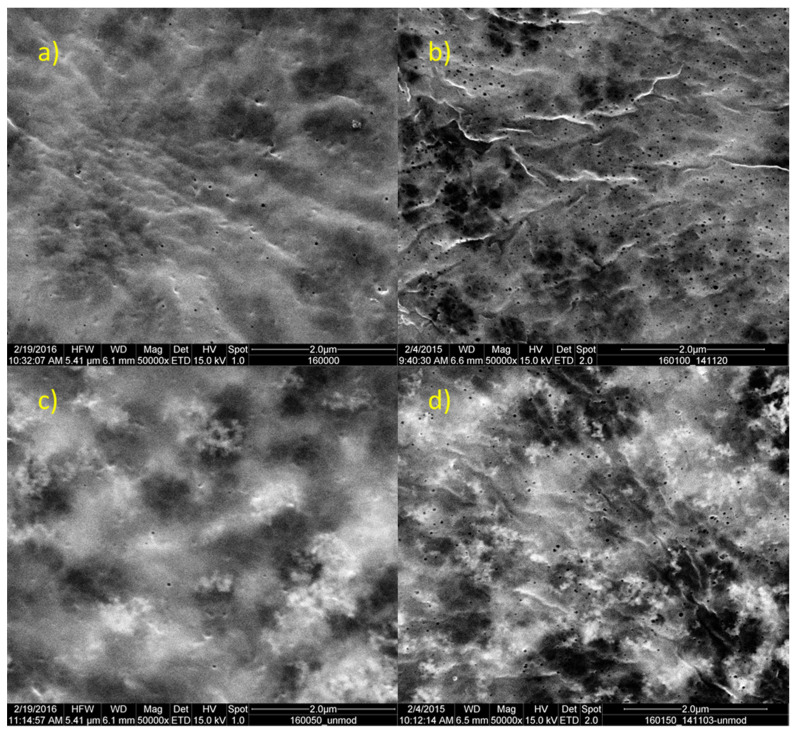
Outer surface morphology of PVDF membranes with and without PVP as well as without and with ZnO nanoparticles. (**a**) 0% PVP, 0% NP; (**b**) 1% PVP, 0% NP; (**c**) 0% PVP, 50% NP; (**d**) 1% PVP, 50% NP; magnification 50,000×.

**Figure 3 membranes-10-00197-f003:**
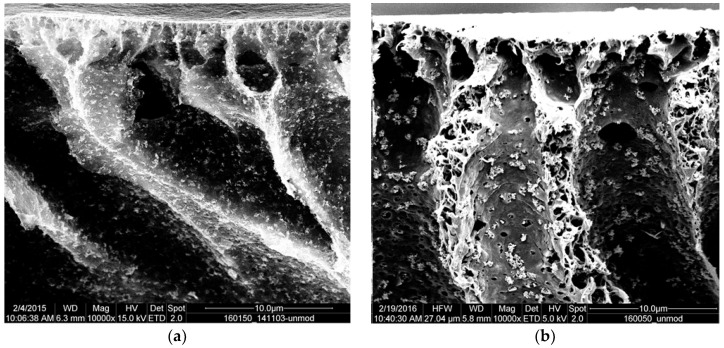
Nanofiller agglomerates in the cross-section of PVDF membranes obtained from dope solutions with 50% *w/w*(PVDF) zinc oxide. (**a**) with 1% PVP; (**b**) without PVP; magnification 10,000×.

**Figure 4 membranes-10-00197-f004:**
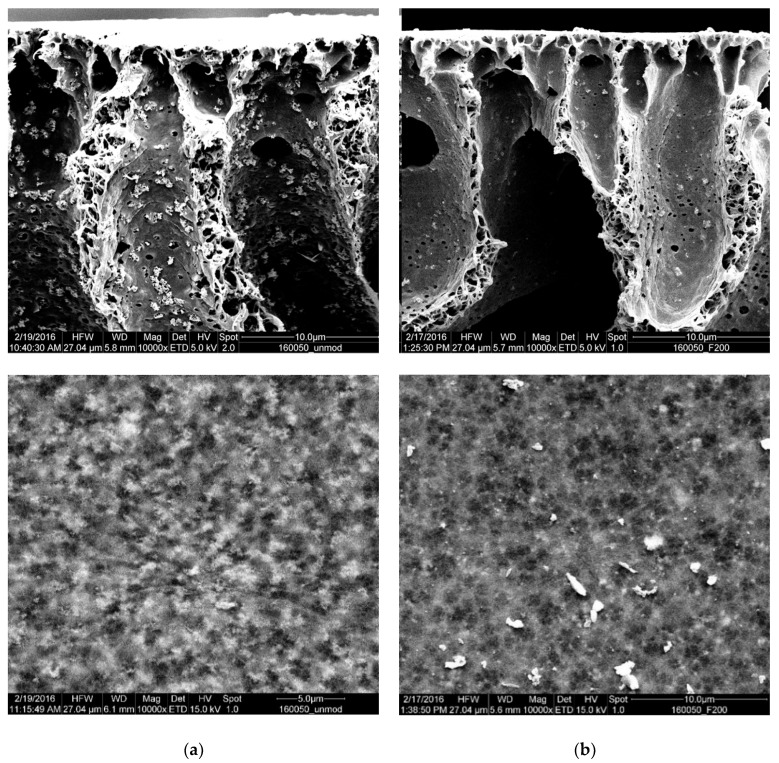
Nanofiller agglomerates in the cross-sections (upper pictures) and on the top layer outer membrane surface (lower pictures) of PVDF membranes obtained from dope solutions without PVP and with 50% *w/w*(PVDF) zinc oxide. (**a**) reference; (**b**) agglomerated; magnification 10,000×.

**Figure 5 membranes-10-00197-f005:**
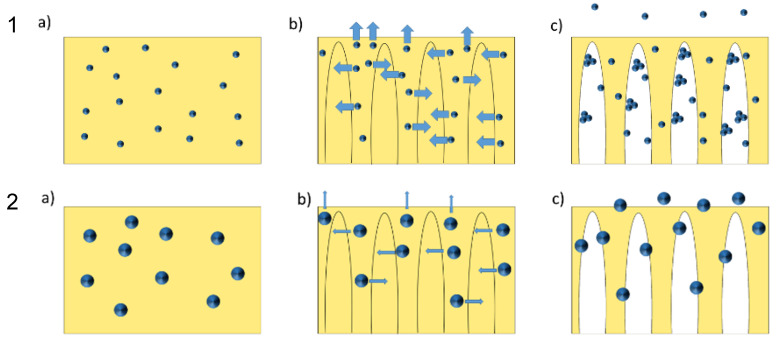
Schematic visualization of the behavior of nanoparticles in the dope solution during the phase separation. Case 1: not aggregated; case 2: aggregated. (**a**) dope solution; (**b**) during NIPS; (**c**) after NIPS).

**Figure 6 membranes-10-00197-f006:**
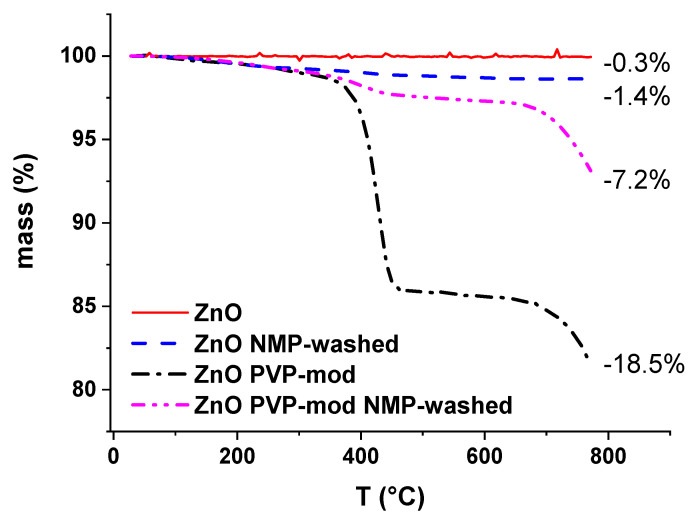
Thermogravimetric analysis of ZnO nanoparticles before and after modification with NMP as well as after washing with NMP for 2 days at 60 °C. Final relative mass difference compared to initial state is also indicated.

**Figure 7 membranes-10-00197-f007:**
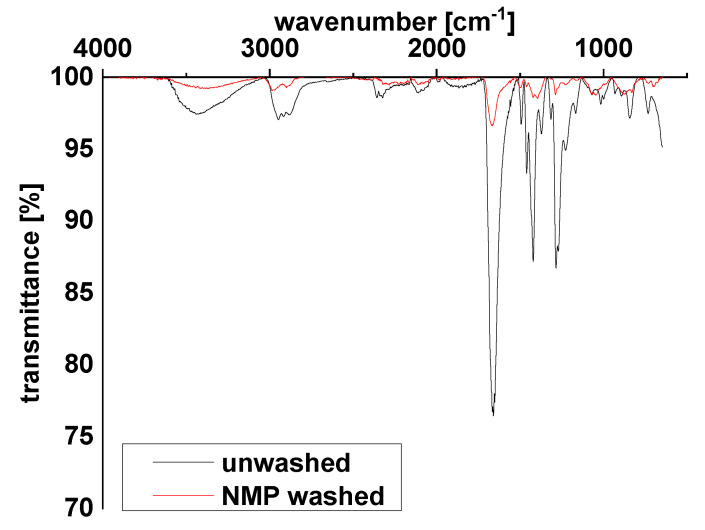
IR spectra of the PVP-modified ZnO nanoparticle before and after extensive washing with NMP.

**Figure 8 membranes-10-00197-f008:**
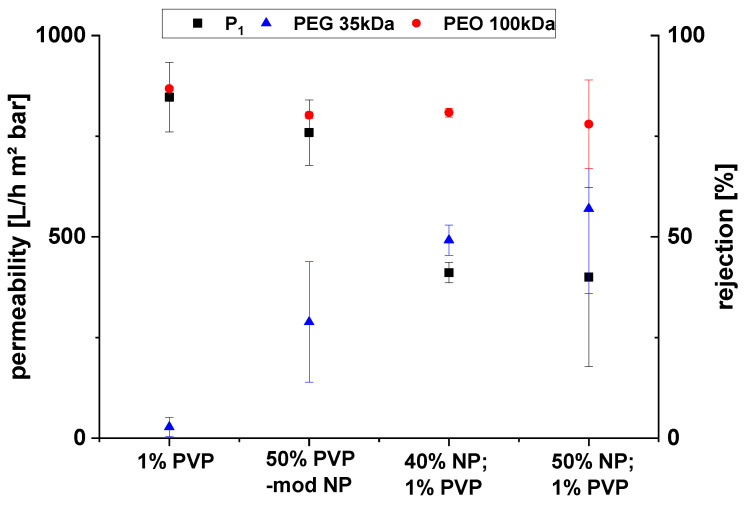
Separation performance comparison for PVDF membranes obtained with PVP-modified ZnO nanoparticles with membranes prepared from dope solutions with same PVP content but no ZnO (“1% PVP”), as well as dope solutions containing either 40% or 50% pristine ZnO and 1% PVP.

**Figure 9 membranes-10-00197-f009:**
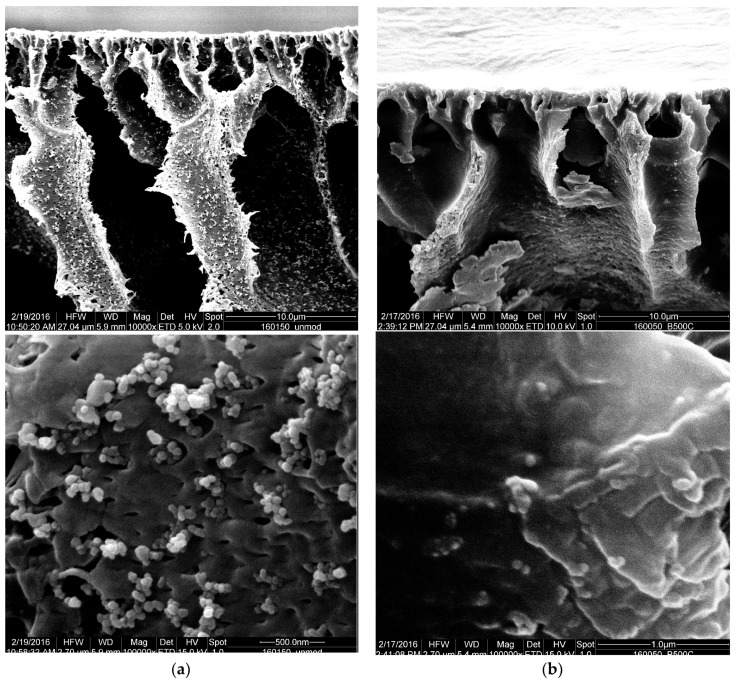
Agglomerates in the cross-section of PVDF nanocomposite membranes prepared with 50% *w/w* (PVDF) nanoparticles. (**a**) reference with unmodified ZnO and 1% PVP; (**b**) only PVP modified ZnO used. Upper pictures: magnification 10,000×; lower pictures: magnification 100,000×.

**Table 1 membranes-10-00197-t001:** Criteria for rating nanoparticle dispersion quality in the solvents used for analysis.

Rating	Criteria
1	d_H_ ≤ 120 nm and stable over at least 3 days
2	d_H_ > 120 nm or size unstable
3	d_H_ > 120 nm and size unstable
4	size steadily increasing
6	unstable

**Table 2 membranes-10-00197-t002:** Ultrafiltration performance of PVDF membranes prepared with and without PVP as well as with and without ZnO nanofiller in the dope solution.

c(PVP)	c(NP)	Water Permeability		Solute Rejection [%]				n
[%w]	[%*w/w*(PVDF)]	[L/h m² bar]	SD	PEG 35 kDa	SD	PEO 100 kDa	SD	
0	0	31	4	65	17	78	7	4
0	50	23	1	76	13	85	3	2
1	0	847	87	3	2	87	0	3
1	50	400	222	57	21	78	11	3

SD—standard deviation; n—number of independent membranes tested.

**Table 3 membranes-10-00197-t003:** Ultrafiltration performance of PVDF membranes prepared without PVP at 50% *w/w* (PVDF) zinc oxide without and with sonication of the nanofiller during dope solution preparation.

Type	Water Permeability	Solute Rejection [%]					N
	[L/h m² bar]	SD	PEG 35 kDa	SD	PEO 100 kDa	SD	
agglomerated	21	0	64	5	70	2	2
reference	23	1	76	13	85	3	2

**Table 4 membranes-10-00197-t004:** Rating of different solvents as dispersants for pristine and PVP-modified ZnO nanoparticles (for criteria see Table 1).

Solvent	Pristine ZnO	PVP Modified ZnO
	Rating	Rating
acetone	6	6
acetonitrile	6	6
2-butanol	2	2
chloroform	6	3
dimethylacetamide	2	2
dimethylformamide	1	2
dimethyl sulfoxide	6	2
1,4-dioxane	3	6
ethanol	1	4
ethyl acetate	2	6
*n*-hexane	6	6
methanol	1	2
methyl ethyl ketone	4	6
*N*-methylpyrrolidone	6	1
dichloromethane	6	3
tetrahydrofuran	2	6
*p*-xylene	6	6

**Table 5 membranes-10-00197-t005:** Hansen solubility parameters for pristine and PVP-modified ZnO nanoparticles obtained from the data shown in Table 4; literature values for PVP and PVDF for comparison.

Material	δ_D_	δ_P_	δ_H_	δ_Tot_	R_o_	fit
	[MPa^1/2^]	[MPa^1/2^]	[MPa^1/2^]	[MPa^1/2^]	[–]	[–]
ZnO	15.8	7.9	16.8	24.4	10.0	1.00
ZnO PVP-mod	19.4	14.3	16.4	29.2	11.3	0.83
PVP [38]	21.4	11.6	21.6	32.5	17.3	-
PVDF [38]	17.0	12.0	10.2	23.2	4.1	-

**Table 6 membranes-10-00197-t006:** Results of the mechanical characterization for various PVDF membranes.

NP Type	c(NP) in Dope	Tensile Strength	SD	Elongation at Break	SD	E Modulus	SD
	[% *w/w* (base Polymer)]	[N]		[%]		[N/mm²]	
None	0	3.66	0.7	43.0	10.5	1.03	0.27
Pristine	50	4.45	0.3	39.4	6.5	1.75	0.14
PVP modified	50	4.12	0.4	63.5	26.8	1.01	0.13

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
