# Peer review of "Polymer Nanocomposite Ultrafiltration Membranes: The Influence of Polymeric Additive, Dispersion Quality and Particle Modification on the Integration of Zinc Oxide Nanoparticles into Polyvinylidene Difluoride Membranes"

_membranes, 2020, doi:10.3390/membranes10090197_

Round 1

Reviewer 1 Report

This study describes the influence of polymeric additive, dispersion quality and particle modification on the integration of zinc oxide nanoparticles (NPs) into polyvinylidene difluoride membranes. Specifically, the NPs dispersion and Hansen solubility parameters were developed to quantify the dispersion and interaction of NPs and modified NPs in the casting solution of mixed matrix membrane. The proposed mechanism is clear to elucidate the behavior of nanoparticles in the casting solution during the phase separation. This is an interesting work and it is well-written. Therefore, I recommend the publication of this work in Membranes a minor revision. The following comments may be considered to further improve the quality of the work.

  1. Please check the format in line 289 and line 397, which shows “ Error! Reference source not found…”
  2. What is the reason for the authors to use high concentration NPs in the casting solution (50% wt/wt PVDF)?
  3. It may be more suitable to use NPs instead of NP for the abbreviation of nanoparticles.
  4. 4 shows the morphology of outer membrane surface of PVDF membranes (PVDF and PVDF/ZnO). The ourter membrane surface might be top surface of membrane. How about the bottom surface of membranes.

Reviewer 2 Report

  1. The author need to mention about the ZnO nanoparticle size.
  2. ZnO was modified with PVP just by physical mixing. There will be clear picture if SEM or TEM image of these modified and unmodified namoparticles are presented.
  3. Molecular weight of PVP is also an important factor. It need to be discussed as well.
  4. In the introduction there is need of discussion on similar kind of work on PVP modification if reported in the literature.
  5. From TGA it is evident that modified ZnO swell in NMP and PVP is water soluble. So how about its stability when used in PVDF nanocomposite membrane.
  6. Cross sectional SEM image are not that clear and if higher magnification image can be added then there might be better understanding about the interaction with the PVDF matrix.

Reviewer 3 Report

This manuscript presented the fabrication of PVDF membranes with ZnO nanoparticles, and tried to draw a conclusive argument that could describe well the effect of nanoparticles and additives/modifiers, an interesting mechanism was proposed. The work is well supported with data in general, while there are still several issues to be addressed by the authors before the manuscript can be accepted for publication.

  1. Line 72, ‘eighteen works can be found…’ is misleading; there are definitely more than 18 papers in literature. The authors may rather put ‘eighteen representative works were selected…’
  2. Line 134, please include the reference for choosing n=2.03.
  3. Please confirm that the rating of 6 in Table 1 is correct. It is kind of unusual to have a missing rating of 5.
  4. Line 249, please be specific the rejection of which solute is increased since that of PEO100k is sort of decreased.
  5. Line 278, the SEM images in Figure 4 is not very clear, so it is hard to conclude that all agglomerates positioned at the pore walls.
  6. Line 294, the rejection values in Table 3 are not virtually the same, judging especially from the SD of PEO100.
  7. It would be very interesting if the authors could try to explain the data in the literature with the mechanism and theories proposed in this study.

Author Response

Please, see attachment
